# An Empirical Analysis of the Impact of Digital Economy on Manufacturing Green and Low-Carbon Transformation under the Dual-Carbon Background in China

**DOI:** 10.3390/ijerph192013192

**Published:** 2022-10-13

**Authors:** Wei Zhang, Hao Zhou, Jie Chen, Zifu Fan

**Affiliations:** School of Economics and Management, Chongqing University of Posts and Telecommunication, Chongqing 400000, China

**Keywords:** digital economy, manufacturing, low carbon, green transformation, mediating effect

## Abstract

The deep integration of digital economy and green development has become an inevitable requirement and an important aid in achieving the goal of carbon peaking and carbon neutrality and promoting high-quality economic development. At the same time, the manufacturing industry is the main sector of energy consumption and carbon emissions in China and the main force for achieving the carbon peaking and carbon neutrality goals. This paper constructs a mathematical model to measure the scale of the digital economy development and the efficiency of the green, low-carbon transformation of the manufacturing industry. It builds a panel data model to study the effect of the development of the digital economy on the green, low-carbon transformation of the manufacturing industry based on data of 30 Chinese provinces from 2016 to 2020. The results find that (1) there is a significant positive effect of the digital economy on the green, low-carbon transformation of the manufacturing industry, with an impact coefficient of 0.477, and this finding remains significant in the robustness test. (2) A further test of the mediating effect finds that the digital economy can drive the green, low-carbon transformation of the manufacturing industry by enhancing technological innovation, and it shows a partial mediating effect that accounts for 28% of the total effect. (3) In the regional heterogeneity analysis, it is found that the effect of the digital economy in promoting manufacturing transformation is more prominent in the central region, and the impact coefficients are 0.684, 0.806, 0.340, and 0.392 for the east, central, west, and northeast regions, respectively. This study can provide a theoretical basis and policy support for governments and enterprises.

## 1. Introduction

China attaches great importance to the development of the digital economy. General Secretary Xi Jinping has repeatedly pointed out that it is necessary to seize the opportunity to develop the digital economy, promote the industrialisation of digital and industrial digitalisation, and promote the deep integration of the digital economy and the real economy. On 14 January 2022, the State Council issued the “Fourteenth Five-Year Plan” for the development of the digital economy, and on 16 January, the magazine “QiuShi” published an important article by General Secretary Xi Jinping, “Continuously strengthen, improve and expand China’s digital economy”, underscoring the country’s increased attention to the digital economy. In September 2020, China proposed to achieve peak carbon dioxide emissions by 2030 and strive to achieve carbon neutrality by 2060. The carbon peaking and carbon neutrality goals were upgraded to a national strategy. By the end of 2021, there were more than 3.5 million enterprises in 31 categories in China’s manufacturing industry, accounting for more than 40% of the total number of enterprises in society. With a strong base and rapid growth, the total energy consumption and carbon emissions of manufacturing enterprises account for two-thirds of the secondary industry and one-third of the total energy consumption and carbon emissions in China, meaning the green and low-carbon transformation of the manufacturing industry is urgent. Therefore, a key question to be answered is whether the digital economy can drive the development of the green and low-carbon transformation of the manufacturing industry. If the logic is valid, we need to clarify the mechanism and effect measurement of the digital economy through which the green and low-carbon transformation of the manufacturing industry are driven, so that a theoretical basis and realistic support for promoting high-quality economic development can be provided.

(1) Research on the digital economy. The term digital economy was first introduced by Tapscott in 1996 [1], and since then, the digital economy has roughly gone through three stages: the information economy, the internet economy, and the new economy [2,3]. However, the definition of the digital economy has been focused on different historical stages, and there is no unified standard [4,5]. Early definitions focused on covering digital technology productivity, emphasising the digital technology industry and its market-oriented applications, such as communication equipment manufacturing, the information technology service industry and the digital content industry [6]. The digital economy is an economic activity with digital information (including data elements) as the key resource, an internet platform as the main information carrier, driven by digital technology innovation, and a series of new models and business models as the manifestation [7]. Given the depth of research, the focus gradually shifted to the interpretation of the economic function of the digital technology and the change in digital technology in terms of production relations [8,9]. The upgrading of traditional industries through the deep integration of digital technologies has become the basic model for the digital transformation of the economy. First, the application of technologies such as the internet and blockchain in the field of production has changed the spatial scope of industries and broken the definition of the traditional industrial boundary [10,11]. Second, the development of digital technology has weakened the role of spatial relationships as a link between enterprises, and the digital transformation of the industry along with the Internet of Things has strengthened industrial synergies and given rise to the networked development of industrial organisations [12]. Third, technologies such as big data, artificial intelligence, and cloud computing provide economists with the possibility of accessing complete information.

(2) Research on the green and low-carbon transformation of the manufacturing industry. The essence of green, low-carbon transformation is harnessing the green, low-carbon economy and high-quality development to achieve sustainable development by saving resources, reducing pollution, and improving production efficiency [13]. Academics mostly determine research methods, indicators, and scales according to the characteristics and perspectives of their respective research fields to measure the efficiency of green transformation in manufacturing and identify the influencing factors [14,15]. (1) The measurement of green and low-carbon transformation in manufacturing, given the rich meaning of green transformation and the many factors involved whereby the integrated index system method and total green factor productivity method are mainly used to comprehensively measure and evaluate green industrial transformation [16,17]. (2) At this stage, the influencing factors of green and low-carbon transformation in the manufacturing industry primarily involve domestic and foreign scholars analysing both internal and external aspects [18,19]. Most studies have been conducted using regression models and spatial econometric models to dissect the factors influencing the development of green transformation from the perspectives of technological innovation, environmental regulation, and resource endowment [20,21,22].

(3) Research on the impact of the digital economy on the development of the manufacturing industry. (1) The digital economy promotes the transformation and upgrading of the manufacturing industry. The digital economy can profoundly affect the traditional manufacturing industry from the demand side, production side, and management side, and the servitisation of manufacturing will become the future direction of manufacturing upgrading [23,24]. (2) The digital economy promotes the development of the manufacturing industry. The digital economy can improve the innovation capacity and service level of the manufacturing industry, optimise and integrate the supply chain and value chain, and improve international competitiveness, thereby promoting the development of the manufacturing industry [25]. (3) The digital economy promotes the industrial restructuring of the manufacturing industry through improving the level of digital infrastructure construction and digital scientific research; additionally, promoting the development of the digital industry can optimise the industrial structure [26]. (4) The digital economy promotes the high-quality development of the manufacturing industry. Strengthening the integration and development of the digital economy and manufacturing industry can promote the high-quality development of the manufacturing industry through synergy and sharing [27,28].

The green and low-carbon transformation of manufacturing industry involves a variety of influencing factors, such as technological innovation, environmental regulation, resource endowment, and government policies. A consensus has been reached in academic circles on the importance of the digital economy on green economic for growth, energy saving and emission reduction in the manufacturing industry, and industrial structure transformation and upgrading. Many scholars believe that technological innovation has a positive role in promoting green transformation in the manufacturing industry [29]. However, the existing research mainly focuses on the development of digital economy, research on green and low-carbon transformation mainly focuses on carbon emissions, and a few scholars have studied how the development of the digital economy affects carbon emissions. The digital economy can positively impact carbon efficiency in the long run by mitigating labour misallocation and capital misallocation, promoting a low-carbon economy by improving the overall efficiency of source allocation, and decreasing carbon emissions by enhancing energy intensity [30,31,32]. The total energy consumption and carbon emissions of manufacturing enterprises account for two-thirds of the secondary industry and one-third of the total energy consumption and carbon emission in China, meaning the green and low-carbon transformation of the manufacturing industry is a practical problem that must be solved. The mechanism of action and the influence path of the digital economy on the green, low-carbon transformation of the manufacturing industry is complex. Scholars have analysed it in terms of industrial structure, green technological innovation, digital total factor productivity, and energy efficiency [33,34]; however, there is still no unified conclusion. There is also a need for more in-depth research on the interaction between the relevant elements of the digital economy and on the green, low-carbon transformation system of the manufacturing industry. This paper constructs a panel data model to study the effect level of the digital economy development on the green, low-carbon transformation of the manufacturing industry to provide a theoretical basis and realistic support for the government to formulate relevant policies.

## 2. Theoretical Analysis

The digital economy plays a vital role in the green and low-carbon development of the manufacturing industry in three main ways: changing production methods, reshaping personnel structure, and improving technological innovation to promote the green and low-carbon development of the manufacturing industry.

(1) The digital economy has adjusted the production methods of the manufacturing industry to become “green” and “efficient”.

With the continuous development of the digital economy, digital information technology represented by big data, the Internet of Things, and cloud computing has transformed the production methods of the manufacturing industry while driving the overall innovation level, and traditional manufacturing industries began to use information technology, internet technology, and industrial intelligence to carry out production activities using digital technology. The digital economy can directly drive high-quality green development, with industrial structure adjustment and green technology innovation being significant mediating mechanisms. The digital economy transforms data and technology into new factors of production and, through deep integration with traditional industries, enables industrial development to integrate new factors of production and improve resource allocation efficiency, thus contributing significantly and positively to high-quality development [35]. Further green production technologies and methods have emerged and been applied to effectively address the problems of low production efficiency and serious production pollution in the traditional manufacturing industry.

(2) The digital economy has reshaped personnel structures within the manufacturing industry in an advanced direction, promoting the green and low-carbon development of the manufacturing industry.

Human capital is the basic condition for the green development of the manufacturing industry driven by the digital economy. The development and popularisation of digital technology bring about changes in the demand for industrial labour skills. Digital economy workers have a high level of education, are young, and have not worked many years [36]. A high proportion of highly skilled workers and high-level talent is needed to achieve the green transformation and upgrading of the manufacturing industry. The introduction of digital economy technology enables employees to conveniently share knowledge and learn from each other, improving personnel knowledge and promoting the advanced employment structure of the labour force. The transformation of the human capital structures of manufacturing enterprises will also inject fresh blood into the green and low-carbon development of the manufacturing industry. Digital technologies are becoming increasingly ubiquitous, and while raising labour efficiency, there are negative impacts on workers who lose employment [37]. Previously, repetitive and simple jobs were replaced by automated machines and computers, and increasingly complex and higher-skilled jobs were needed. Today, technology, learning, and knowledge are becoming increasingly important to the development of the manufacturing industry; middle- and low-skilled labour will be gradually replaced, while high-level labour with higher education or specific skills will gain greater advantages. High-quality human resources have an important role in the green and low-carbon transformation of manufacturing.

(3) The digital economy promotes the green development of the manufacturing industry by enhancing technological innovation capabilities.

Green development driven by innovation is the key to achieving the transformation and upgrading of pollution-intensive industries and improving economic quality and efficiency [38]. In a sense, digital technology is the core of the digital economy and the main engine of its development. Similarly, technological innovation is an important driving force for the green development of the manufacturing industry. The high permeability of the digital economy enables new digital technologies to be rapidly and widely applied to the manufacturing industry, promoting manufacturing technology innovation and industrial innovation, realising the digital and green transformation of the manufacturing industry, and helping China’s manufacturing industry eliminate the development modes of high pollution, high consumption, and high emission.

## 3. Measurement of the Digital Economy

### 3.1. Evaluation Index

In this paper, based on existing research and combined with actual research needs, an index system for the digital economy development level evaluation is constructed based on 15 indicators in four dimensions—basic indicators, industrial indicators, environmental indicators, and integration indicators—with the objective of analysing and studying the development level of the digital economy of 30 provinces from 2016 to 2020. As shown in Table 1 below, the principles of the scientific selection of indicators, data availability, continuity, and the representativeness of indicators are followed.

(1) Basic indicators. The development of the digital economy is closely related to the internet and information infrastructure. This paper selects the number of mobile IPV4 addresses, the base stations of mobile phones, the lengths of optical cable lines, broadband subscriber internet ports, and the popularisation rate of mobile phones to reflect the construction level of the digital infrastructure [39,40].

(2) Industry indicators. The development of the digital economy is also closely related to the development of high-end frontier technologies. This paper selects the business volume of telecommunications services, income from related software businesses, and the number of top 100 internet companies to reflect the development scale of the digital industry, encompassing the telecommunication business, software industry, and internet development.

(3) Environmental indicators. The number of software developers, invention patent applications, full-time equivalents of R&D personnel, and transaction value in the technical market were selected. The number of software developers and the full-time equivalents of R&D personnel reflect the input of science and technology innovation talent, and the number of invention patent applications and transaction value in technical market reflect the intensity of science and technology innovation in each province.

(4) Convergence indicators. This indicator is used to measure the degree of integration of provinces with the digital economy. Sales and purchases through e-commerce and the proportion with e-commerce transaction enterprises are selected to reflect the consumption level of enterprises and residents on the internet as a platform [41]. The China Digital Financial Inclusion Index is used to measure digital finance development.

### 3.2. Data

The years 2016–2020 are selected as the study period, and the sample comprises 30 provinces, excluding Tibet (which is missing data). The data are mainly obtained from the China Statistical Yearbook, an online public information collation, and the China Information Industry Yearbook. The missing data are processed as follows: interpolation and analogy methods are adopted to supplement the missing data. The balanced panel data of 30 provinces from 2016 to 2020 are finally obtained through data collection and processing.

### 3.3. Result

(1) Each province

Let $v_{\mathrm{ij}}\;$be the original data of the *j*-th indicator in the *i*-th evaluation object (*i* = 1, 2, 3, …, *n*; *j* = 1, 2, 3, …, *m*). In order to make the data of different calibres comparable and eliminate the difference of the dimension between the indicator data, the original data are standardised; if the indicator is a positive indicator, then the processing formula can be written as follows:(1)$$x_{ij}=\frac{v_{ij}-\min\{v_{j}\}}{\max\{v_{j}\}-\min\{v_{j}\}}+0.0001$$

If the indicator is negative, the treatment formula can be written as follows:(2)$$x_{ij}=\frac{\max\{v_{j}\}-v_{ij}}{\max\{v_{j}\}-\min\{v_{j}\}}+0.0001$$

$v_{ij}$ (*i* = 1, 2, 3, …, *n*; *j* = 1, 2, 3, …, *m*) is the raw data of the *j*-th index of the *i*-th evaluation object and is the dimensionless value after standardisation.

Let $p_{ij}$ be the weight of the *j*-th indicator in the *i*-th evaluation object, $e_{j}$ be the entropy value of the *j*-th indicator, and $r_{j}$ be the coefficient of variation in the *j*-th indicator, where the number of evaluation objects is the weight of the first evaluation indicator (*j* = 1, 2, 3, …, *m*). Each coefficient is calculated by the following formulas.
(3)$$p_{ij}=\frac{X_{ij}}{\sum_{i=1}^{n}X_{ij}}$$
(4)$$e_{j}=-\frac{1}{lnn}\sum_{i=1}^{n}p_{ij}lnp_{ij}$$
(5)$$r_{j}=1-e_{j}$$
(6)$$w_{j}=\frac{r_{j}}{\sum_{j=1}^{m}r_{j}}$$

According to the weights of each indicator calculated by Equation (6) and the data of each indicator after standardisation, the level of digital economy development can be calculated by Equation (7), where DE represents the digital economy score, and $w_{j}$ and $x_{j}$ represent the weight and standardised value, respectively.
(7)$$DE=\sum_{j=1}^{m}w_{j}x_{j}$$

Table 2 shows the estimated results of the comprehensive index of the digital economy development level from 2016 to 2020. It can be seen that the digital economy development level has significant heterogeneity in time and space. First, the mean value of China’s overall digital economy development level grew from 0.1174 in 2016 to 0.1917 in 2020, with an average annual growth rate of 13.09%, but the number of cities reaching the average level was less than one-third, among which, 9 provinces and cities were above the mean value in 2016, 10 in 2017, 8 in 2018, 8 in 2019, and 8 in 2020. Second, from the perspective of the provinces, the development of the head team remained stable; specifically, Guangdong, Beijing, Jiangsu, Zhejiang, Shanghai, Shandong, and other eastern coastal regions of the digital economy development level were consistently in the leading positions. Guangdong ranked first for four years except in the year 2017, which may have been due to the national policy first being established in the east, where the electronic information class of investment is stronger, and the digital infrastructure is more advanced. Since the eastern provinces are China’s early industrial bases, the development model is heavy on industrialisation and it lacks digital resources, making the development of the digital economy more difficult. Other cities have also improved significantly, with Jiangxi, Xinjiang, Guizhou, Inner Mongolia, Jilin, and other provinces having the highest average annual growth rates, all exceeding 20%. Although some of the lower levels of development of the provinces are trying to narrow the gap with the head team, it is undeniable that the interprovincial gap is still more pronounced. For example, in 2020, the digital economy of Guangdong in the eastern region (0.7380) was 27 times that of Qinghai in the four regions (0.0274) and 17 times that of Hainan in the eastern region (0.0427). The provinces of Qinghai and Ningxia have a relatively backwards level of economic development: they had a late start in digital economy development, whereby the number of cell phone base stations is the lowest in the country, and the average annual number of software development employees in Qinghai in 2016–2020 was less than 100. The backwardness of the digital infrastructure and the lack of human capital further limit the development of the digital economy and will have an impact on their economy and will widen the digital divide between cities.

(2) The four major regions

Although the level of digital economy development is accelerating in all of China’s regions, the digital economy development gap between the eastern region and the central and western regions is significant and continues to widen, as shown in Figure 1. The eastern region is leapfrogging ahead of the other regions; the western region is growing steadily with the fastest average annual growth rate of 16.93%; the central region is ahead of the western region and the northeast region with a more significant advantage, and an average annual growth rate of 15.83%; furthermore, the northeast region had almost the same level of development in 2017 and 2018 and insignificant growth in 2019 and 2020; indeed, its average annual growth rate was the lowest among all regions. The digital economy development level in 2020 in the eastern region was 0.3571, 0.1477 in the central region, 0.0934 in the western region, and 0.0892 in the northeast region; the western region figure is very close, which is perhaps because the digital economy foundation in the eastern region is better and the digital-economy-level stock is large, while the digital economy development level in other regions is low. Therefore, the leading growth rate is also reasonable.

## 4. Measurement of the Manufacturing Transformation Efficiency

### 4.1. Evaluation Index

In this paper, the four dimensions of economic efficiency, green development, technological innovation, and digital integration are selected, with a total of eight indicators, as shown in Table 3. Economic efficiency represents the business development of the manufacturing industry, and the total profits of industrial enterprises above a designated size (manufacturing industry) are selected for measurement. Green development mainly reflects clean production, resource utilisation, and friendliness to the ecological environment of the manufacturing industry under the new concept of green development. The energy consumption per unit of industrial value added, industrial wastewater discharge, total industrial wastewater emissions, total industrial waste gas emissions, and common industrial solid waste were used to measure the level of green development [42,43]. To take the green, low-carbon transformation path, the manufacturing industry must improve its green technology innovation capability and must increase innovation inputs to achieve a higher level of innovation output. The R&D expenditure and the number of valid invention patents of the industrial enterprises of the above-designated size are selected to measure the level of technology innovation. With informatisation driving industrialisation and industrialisation promoting informatisation, the integration of both is necessary for the green and low-carbon development of the manufacturing industry. The data of the digital economy development level obtained from the previous measurement are used as the measurement index of the digital integration application in the manufacturing industry.

### 4.2. Data

The years 2016–2020 are selected as the study period, and the sample comprises 30 provinces, excluding Tibet (which is missing data). The data are mainly obtained from the China Statistical Yearbook, the provincial and municipal statistical yearbooks, an online public information collation, and the China Information Industry Yearbook. The missing data are processed as follows: interpolation and analogy methods are adopted to supplement the missing data. The balanced panel data of 30 provinces from 2016 to 2020 are finally obtained through data collection and processing.

### 4.3. Result

(1) Each province

The calculation method here is consistent with DE and will not be repeated.

Table 4 shows the estimated results of the green, low-carbon transition efficiency of the manufacturing industries in each province from 2016 to 2020. It can be seen that the green, low-carbon transition efficiency of manufacturing industries in each province has significant heterogeneity over time. First, in China as a whole, the mean value of the green, low-carbon transition efficiency of China’s manufacturing industry increased from 0.0887 in 2016 to 0.1917 in 2020, with an average annual growth rate of 6.64%, but the number of cities that reached the national average was less than one-third, with four provinces and cities above the mean in 2016, five in 2017, six in 2018, seven in 2019, and six in 2020. Second, from the perspective of provinces, the development of the leading places remained stable; specifically, Shandong, Beijing, Guangdong, Jiangsu, Hunan, and Shanghai were in the leading positions of transformation efficiency year-round, with Shandong ranking first for five consecutive years. Other provinces and cities also significantly improved their transformation efficiency, with Shanxi, Hebei, Jiangxi, Anhui, Hunan, and other provinces ranked in the top positions in terms of average annual growth rate, with Shanxi ranking first by a huge margin. The transformation efficiency of Shandong, which ranks first, has been decreasing annually, and the gap between Shandong and the other provinces has been gradually narrowing, in which the difference with the second place decreased from 0.47 in 2016 to 0.14 in 2020. Although the less efficient provinces are trying to narrow the gap with the head team, it is undeniable that the interprovincial gap remains significant. For example, the transformation efficiency of Shandong in 2020 (0.4136) was 12 times higher than that of Gansu (0.0347) and 9 times higher than that of Jilin (0.0439).

Table 5 shows the variables and explanations. MTE and DE are the explained variable and core explanatory variable, and are obtained from chapter 3 and 4. Other variables are from public data.

(2) The four major regions

As shown in Figure 2, the transformation efficiency of the four regions generally shows a year-on-year increase, with the eastern region still far ahead of the other regions. The central region has the largest span of development efficiency, with the fastest average annual growth rate of 14.94%. Although the western region has a lower level of development efficiency, it continues to grow, with an average annual growth rate of 8.73%. The northeast region has a slightly fluctuating development trend compared with the western region but still ranks third with an average annual growth rate of 7.53%, while the eastern region ranks last with an average annual growth rate of 3.56%. In terms of the efficiency of the green and low-carbon transformation of the manufacturing industry, the manufacturing transition efficiency in the eastern region in 2020 was 0.1927, 0.1266 in the central region, 0.0550 in the western region, and 0.0580 in the northeastern and western regions. This was perhaps due to the higher level of economic development in the eastern region, its earlier stage of transformation, and the more successful digital transformation of enterprise processes and production activities using digital technology. Thus, the efficiency of the green, low-carbon transformation of the manufacturing industry in the eastern region is higher, which can be supported by the results of the digital economy development level measurement in the previous section.

## 5. Study Design

### 5.1. Theoretical Mechanism and Research Hypothesis

(1) Direct influence mechanism

Digital technology is the most important breakthrough technological innovation in the era of the digital economy, and it is also the core driving force of the digital economy for empowering the green development of the manufacturing industry. Digital technology has natural green attributes. Different from conventional economy, outputs from digital economy are more environmentally friendly, with much less energy consumption and environmental emissions, which enhances economic efficiency and reduces energy consumption and environmental pollution, such as sensor technology, which realises the real-time monitoring and intelligent control of the production process, thus effectively reducing pollution emissions. The new generation of information technology is used to research and innovate in the areas of energy savings and emission reductions, flexible manufacturing, biological manufacturing, and other green technologies. Moreover, it is used to transform high energy consumption and high pollution production technologies and processes; reduce the energy consumption of resources in all aspects of production activities; reduce carbon emissions; promote the transformation and development of enterprises in a greener, more economical, and efficient way; and continuously promote the green transformation and upgrading of industries. Based on this, this paper puts forward the following hypothesis:

**Hypothesis** **1.**
*The digital economy drives the green and low-carbon transformation of the manufacturing industry.*


(2) Indirect influence mechanism

Technological innovation is the fundamental way to achieve green development under the constraints of environmental regulation. The digital economy is the most innovative contemporary economic form, and its development has a catalytic effect on innovation. The digital economy promotes the transformation and upgrading of the manufacturing industry by enhancing technological innovation, which is different from the traditional crude development model in which only tangible factor inputs such as labour and capital and tangible economic outputs are considered. Under the green development model, resource factor inputs and undesired outputs such as environmental pollution should be taken into account, and the maximisation of economic benefits and environmental protection should be pursued. The rapid development of the digital economy can squeeze traditional high-pollution and high-energy-consuming industries, promote enterprises to carry out green technology research and development, and realise low carbon production in the manufacturing industry and help China’s manufacturing industry eliminate the development mode of high pollution, high consumption, and high emissions. Thus, this paper puts forward the following hypothesis:

**Hypothesis** **2.**
*The digital economy promotes the development of the green transformation of the manufacturing industry by improving the technological innovation capability of enterprises.*


### 5.2. Model

Firstly, we perform a benchmark regression that does not take into account technological innovation:(8)$$ln(MTEit)=\alpha_{0}+\alpha_{1}ln(DE_{it})+\alpha_{2}ln(SCA_{it})+\alpha_{3}ln(COST_{it})+\alpha_{4}ln(LAB_{it})+\varepsilon_{it}$$

Secondly, we perform a regression of the mediating variables:(9)$$ln(RD_{it})=\beta_{0}+\beta_{1}ln(DE_{it})+\beta_{2}ln(SCA_{it})+\beta_{3}ln(COST_{it})+\beta_{4}ln(LAB_{it})+\varepsilon_{it}$$

Finally, we examine the mechanism path of technological innovation as a mediator variable:(10)$$ln(MTEit)=\gamma_{0}+\gamma_{1}ln(DE_{it})+\gamma_{2}ln(RD_{it})+\gamma_{3}ln(SCA_{it})+\gamma_{4}ln(COST_{it})+\gamma_{5}ln(LAB_{it})+\varepsilon_{it}$$

i represents the province; t represents the year; coefficient $\alpha_{1}$ is the total effect of the digital economy on the green, low-carbon transformation of manufacturing; coefficient $\beta_{1}$ is the effect of the digital economy on technological innovation; coefficient $\gamma_{2}$ is the effect of the technological innovation mediating variable on the green, low-carbon transformation of manufacturing after controlling for the effect of the digital economy; coefficient $\gamma_{1}$ is the direct effect of the digital economy on the green, low-carbon transformation of manufacturing after controlling for the effect of the technological innovation mediating variable; $\alpha_{0}$, $\beta_{0}$, and $\gamma_{0}$ are the constant terms; $\varepsilon_{it}$ is the random error term.

The intermediation effect is equal to the indirect effect and to the $\beta_{1}*\gamma_{2}$, which is related to the total effect and the direct effect as follows:(11)$$\alpha_{1}=\gamma_{1}+\beta_{1}*\gamma_{2}$$

The explanatory variable is the efficiency of green, low-carbon transition in manufacturing, which is denoted by $MTE_{it}$. The core explanatory variable is the level of digital economy development, which is denoted by $DE_{it}$. Based on the literature, this paper argues that there is a mediating effect of technological innovation on the green, low-carbon transformation of the manufacturing industry and selects the R&D expenditure of industrial enterprises above a designated size. To weaken the analysis errors that may be caused by the omission of variables, this paper selects enterprise size, operating cost, and human capital as control variables and selects the proportion of industrial value added to GDP, the main operating cost of industrial enterprises above the scale, and the index of average years of education of the population above 6 years old to represent them.

(3) Data

To minimise the impact on the regression results due to the excessive fluctuation of data values and make the order of magnitude of each variable closer, this paper performs natural logarithmic processing on the indicators of the digital economy development level, technological innovation, industrial scale, and operating costs. The data are obtained from the National Bureau of Statistics, the China Industrial Statistical Yearbook, and the collation of public information on the internet. Due to the extensive lack of data in the province of Tibet, 30 provinces in China are selected as the research objects, and the sample data are balanced panel data from 2016 to 2020. The descriptive statistics of the variables are as follows. The results of the multicollinearity test show that the VIF values for DE, COST, SCA, and LAB are 4.07, 3.89, 1.58, and 1.54. The average VIF value is 2.77, and there is no significant multicollinearity. Table 6 shows the statistical description of the variables. The standard deviation of DE is greater than the standard deviation of MTE. This indicates that the difference in the level of digital economy development among the 30 provinces is greater than the level of manufacturing transformation efficiency.

### 5.3. Result and Analysis

(1) Benchmark regression

The article is analysed using Stata, and the Hausman test results show that all models take the best and most robust estimation results of the fixed effects model. Table 7 shows the results of the benchmark regression of the digital economy on the efficiency of green, low-carbon transformation in manufacturing, and all models have good overall explanatory validity overall. Column (1) is the regression result without adding control variables, and the regression coefficient of the digital economy is 0.559, which is significantly positive at the 1% significance level. Column (2) is the regression result with added control variables. After controlling for enterprise size, operating cost, and human capital level, the regression coefficient of the digital economy is 0.477, which is also significantly positive at the 1% significance level. For every 1% increase in the development level of the digital economy, the efficiency of the green, low-carbon transformation of regional manufacturing industries increases by 0.477%, implying that the development of the digital economy will have a positive impact on the efficiency of the green, low-carbon transformation of manufacturing industries. From the perspective of the control variables, the regression coefficients of the operating costs and human capital are positive, indicating that the increase in operating costs and the improvement in the level of human capital provide strong support for the development of green, low-carbon transformation in manufacturing; the regression coefficient of the enterprise size is negative and insignificant at −0.081, indicating that larger enterprises have a smaller inhibitory effect on the development of green, low-carbon transformation in manufacturing, probably because the larger the scale of enterprises, the more capital and time they need to invest, the longer the transformation cycle takes, and the more difficult the transformation process is.

Regression coefficients of the digital economy are positive and significant at the 1% significance level with or without control variables, and the sign of the correlation of variables derived from the benchmark regression model is consistent with the expected assumptions, indicating that the level of development of the digital economy and the green, low-carbon transformation of China’s manufacturing industry is positively correlated; thus, Hypothesis 1 holds.

## 6. Discussion

### 6.1. Mediation Effect Test

To test whether H2 holds and to determine the share of mediating effects of the technological innovation, this paper conducts a mediating effects test according to the mediating effects test procedure [44], as shown in Figure 3, which is followed for all the mediating effects tests in this paper, and the mediating effects model and test flow chart are as follows:

Table 8 shows the results of the intermediary effect test of the digital economy influencing the development of the green and low-carbon transformation of the manufacturing industry. The results show that there is a partial mediating effect of technological innovation on the digital economy, influencing the green and low-carbon transformation of the manufacturing industry, and the percentage of the mediating effect is 0.281. In Column (1), the influence coefficient of the digital economy is 0.452, which is significantly positive at the 1% significance level, verifying that the digital economy has a positive influence on the green economy. In Column (2), the impact coefficient of the digital economy is 0.516, which is also significant and positive at the 1% significance level, indicating that the digital economy has a significant positive effect on technological innovation and verifying that the digital economy enhances the technological innovation mediating variable capability, keeping other conditions constant. In Column (3), where the mediating variable is added, the coefficient of the impact of the digital economy on manufacturing transformation decreases from 0.452 to 0.331, which is also significantly positive at the 1% significance level, and the change in this coefficient indicates that the improvement of technological innovation capability plays a mediating role. For every 1% increase in technological innovation, the efficiency of green, low-carbon transformation in manufacturing increases by 0.246 units, controlling for other conditions. This is because the most significant feature of the digital economy is that innovative technologies such as big data, cloud computing, and artificial intelligence bring a huge boost to technological innovation in the traditional manufacturing industry, which ultimately boosts the green, low-carbon transformation and upgrading of manufacturing enterprises; thus, Hypothesis H2 is verified.

### 6.2. Heterogeneity Test

Since there are large differences among Chinese provinces in various aspects, the digital economy and the development of green, low-carbon transformation in manufacturing industries show significant heterogeneity in spatial distribution; therefore, it is of practical significance to analyse the impact of the digital economy on green, low-carbon development in manufacturing industries from a regional perspective. For testing purposes, this paper divides China into four regions, namely, the eastern, central, western, and northeastern regions. The regional heterogeneity test results are shown in Table 9. It can be seen that there is a statistically significant positive correlation between the level of digital economy development and the efficiency of green, low-carbon transformation in manufacturing in all four regions. The above results show that the digital economy significantly promotes the development of green, low-carbon transformation in manufacturing in different regions, and that the driving effect of the digital economy in the central region on the development of green, low-carbon transformation in manufacturing is stronger than in the eastern region. This is perhaps because the eastern region has a higher level of digital economy development and a higher level of green, low-carbon transformation in manufacturing, because it started earlier than in other regions and is more efficient. In recent years, the transformation efficiency of the manufacturing industry has been relatively stable, and the average annual growth rate has been low, while the central region, which ranks second in the development level of the digital economy, is ranked first because of the better development of the digital economy in recent years, and the transformation efficiency of its manufacturing industry is increasing linearly. During the “12th Five-Year Plan” period, 95.6% of China’s newly identified reserves were in the west, concentrated in the northwestern provinces, including Inner Mongolia, Shaanxi, Ningxia, and Xinjiang. In 2020, of the top five provinces and municipalities in the energy contribution rate ranking, four were in the western region due to its rich energy resources. The digital economy is not as influential as in the other three regions because of its rich energy resources and the need to protect and thereby guarantee China’s energy supply, hence leading to the large proportion of high-energy-consumption, high-pollution, and high-emission industries.

### 6.3. Robustness Test

To verify the reliability of the results, the following method was used for robustness testing, and the results are shown in Table 10.

(1) Substitution of core explanatory variables

The “Peking University Digital Financial Inclusive Index” is selected as a proxy variable to measure the development level of the digital economy for robustness testing, and the results are shown in Column (1) of Table 10.

(2) The stepwise addition of control variables was used to perform robustness tests to determine whether the fixed-effects model of the basic regression has endogeneity problems caused by omitted variables or two-way causality. The results are presented in Columns (2)–(5) of the table below; it can be seen that the regression coefficients of the core explanatory variable digital economy development level with different combinations of control variables are significant at the 1% significance level, indicating that digital economy development has a significant and stable positive impact on the green, low-carbon transformation of the manufacturing industry.

(3) Delete some samples

The COVID-19 epidemic in 2020 will have had some impact on the development of the digital economy, based on which, the crisis factor has been excluded in this paper, and the results are shown in Column (6). After the partial exclusion of the sample, the positive contribution of the digital economy to the development of the green and low-carbon transformation of the manufacturing industry still holds.

(4) Endogeneity issues

Endogeneity issues may lead to bias in the estimated coefficients of the benchmark regression. Generally, the higher the level of the development of the digital economy is, the more beneficial it is to the green development of the manufacturing industry in that region; however, the region with a higher level of green transformation of the manufacturing industry is, likewise, more likely to attract leading digital economy enterprises, which in turn has a catalytic effect on the local digital economy level. Therefore, there may be an endogeneity problem between the digital economy and the green development of the manufacturing industry due to reverse causality, and this paper adopts the following two methods to test the endogeneity:(1)The core explanatory variable digital economy development level is analysed with a one-period lag, and the results are presented in Column (7) of Table 10. The regression coefficient of digital economy development in period t-1 is significantly positive at the 1% level, which again confirms the robustness of the benchmark regression results.(2)Instrumental variable method

Although this paper controls for relevant variables as much as possible, there may still be omitted variables that are not observed but have an impact on the green, low-carbon development in manufacturing, leading to omitted variable bias in the regression results. To control for the endogeneity problem due to reverse causality and omitted variables, this paper adapts the most commonly used instrumental variables approach [45,46,47,48]; however, considering the limitations of data acquisition, the number of telephones per 100 people in 1984 in each province is selected as the instrumental variable for the digital economy (IV) to alleviate the possible endogeneity problem in the model [49]. This is because the digital economy is closely related to internet technology, while the internet started from the popularisation of landline telephones, and the regions with greater historical penetration of landline telephones are most likely to be the regions with a higher level of penetration of internet pervasiveness. On the other hand, traditional telecommunication tools such as landline phones have almost no influence on the level of green development in the current stage, so there is no significant correlation between them and the disturbance terms affecting the green development of the manufacturing industry, which satisfies the homogeneity requirement of selecting instrumental variables to satisfy the conditions of instrumental variable selection. Because the data sample in this paper is balanced panel data and the number of fixed telephone calls per 100 people at the city level in 1984 is cross-sectional data that does not change over time, their use as instrumental variables does not meet the requirements of panel data for data heterogeneity. For this reason, this paper constructs the number of fixed telephone calls per 100 people (related to individual changes) in 1984 in each city and the interaction term of national IT service revenue (time-related) in the previous year as instrumental variables [50]. The results are shown in Column (8), and they demonstrate that the significance of the control variables varies significantly, but the role of the digital economy in the green development of the manufacturing industry is still significant.

(5) Study Comparison

Compared with past papers, a similarity between them is that we both verify that the digital economy has a significant inhibitory effect on carbon emissions and can promote green and low-carbon development [51,52]. There are also studies contrary to the findings of this paper, in which it was shown that the digital economy can reduce carbon emissions up to a certain range and increase them beyond that [53]. In terms of impact mechanisms, environmental governance, technological innovation, and industrial structure upgrading can indirectly act on the impact of the digital economy on carbon emissions [54]. Few articles have examined the manufacturing industry as an object of study to investigate the impact of the digital economy on its green and low-carbon transformation. The difference in this paper is that we take the manufacturing industry as the research object and study the impact of digital economy on the green and low carbon transformation of the manufacturing industry.

## 7. Conclusions and Policy Implications

This paper starts from the digital economy, based on balanced panel data of 30 provinces in China from 2016 to 2020, and uses fixed effect and mediating effect models to empirically analyse and study the direct and mediating effects of the digital economy on the green and low-carbon transformation of the manufacturing industry and its mechanism of action. The results of the study show the following:

(1) There are significant geographical differences in the development of China’s digital economy. The measurement results show an unbalanced distribution, with the eastern region leading development, followed by the central region and the southwestern and northeastern regions at a lower level of development.

Local governments in the western and northeastern regions should recognise the current situation of regional development and adapt measures to local conditions, which are important prerequisites for the development of the digital economy. They should increase financial support to solidify the construction of the digital economy infrastructure; improve traditional infrastructures such as broadband access ports and cell phone base stations; accelerate the application of emerging technologies such as robotics, big data, and 5G; focus on building digital infrastructure represented by artificial intelligence, industrial internet, 5G base stations, and big data centres; accelerate the cross-fertilisation of new-generation internet information technology with the manufacturing industry; and guide enterprises to reasonably apply digital technology in production activities in an orderly manner. A series of preferential policies can be introduced to encourage traditional manufacturing industries to conduct R&D in digital technology; in particular, high-tech enterprises should be given incentives and loan interest subsidies for key industries and projects.

(2) The development of the digital economy has an overall positive effect on the development of the green and low-carbon transformation of the manufacturing industry and has become one of the most important driving forces of the green and high-quality development of Chinese industry. Additionally, the central region has significantly better digital economy benefits than the other three major regions.

For the further development of China’s digital economy, the Chinese government should fully support the digital economy, play a guiding and supporting role, establish a platform for the development of the digital economy so that information technology can quickly penetrate and be applied to various industries, create a favourable business environment, further attract foreign investment, draw on advanced foreign digital technologies and ideas, give full play to the spillover effects of knowledge and technology, and further release the dividends of the digital economy. In addition, we will lead the digital economy to a higher level of development. Meanwhile, we will accelerate the digital transformation of the whole society and the whole industry and promote regional green and low-carbon development. In the process of promoting the green and low-carbon transformation of the manufacturing industry, we should avoid homogeneous and monolithic development strategies, the western and northeastern regions should be tilted towards the principle of adapting to local conditions, striving to use the development of the digital economy as an opportunity to guide the coordinated development of all regions.

(3) Technological innovation plays an intermediary role, and the digital economy promotes the development of the green, low-carbon transformation of manufacturing industry by improving technological innovation, and the intermediary effect of technological innovation accounts for 28%.

It should increase breakthroughs in vital technologies in key areas such as artificial intelligence, digital platforms, and information security; strengthen the training of talent in this field; improve the education system; and bring into play the independent technological innovation of talents. Furthermore, it should strengthen the core digital technology supply, accelerate the overall layout of green technology innovation, and deeply promote the integration and development of digital technology and green technology. We should also actively explore the application scenarios of digital technology in the process of the green and low-carbon transformation of manufacturing industry; accelerate the deep integration of digital technology with various fields such as new energy development, clean technology, green manufacturing, low-carbon development, pollution control, and resource recycling; and continuously improve the digital content in green technology.

## Figures and Tables

**Figure 1 ijerph-19-13192-f001:**
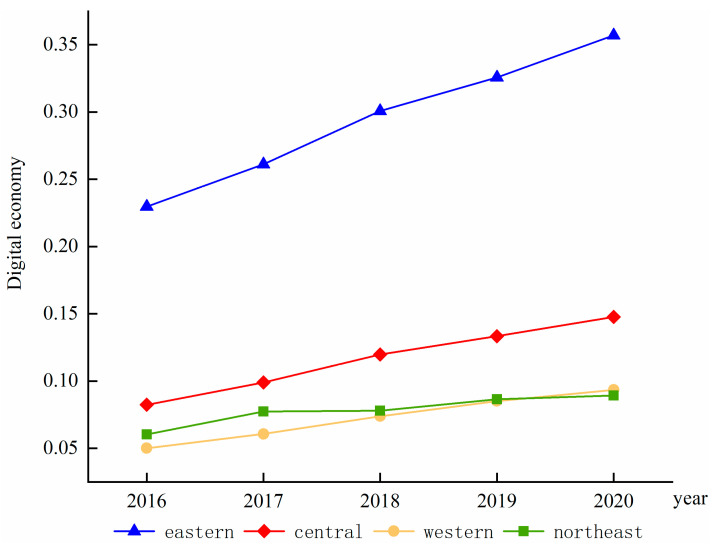
The development level trend of the digital economy in four regions.

**Figure 2 ijerph-19-13192-f002:**
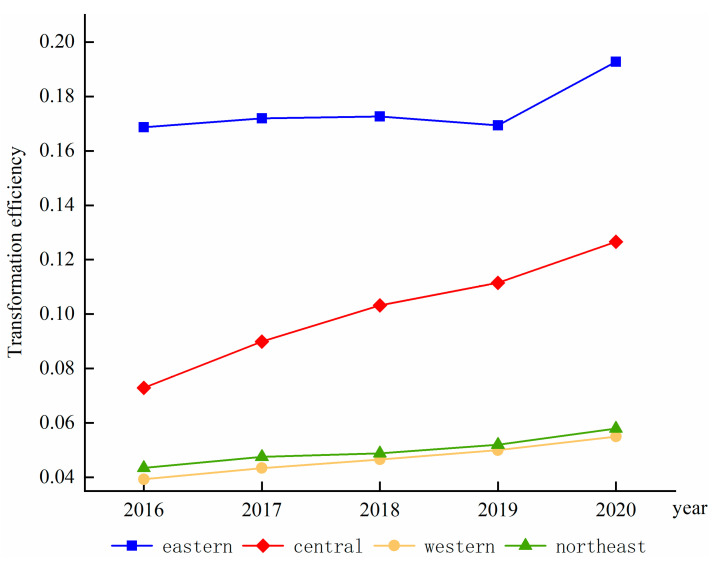
Manufacturing transformation efficiency trends in four major regions.

**Figure 3 ijerph-19-13192-f003:**
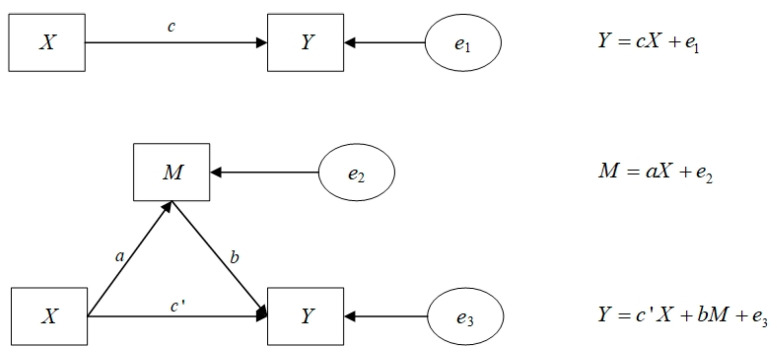
Mediation effect model.

**Table 1 ijerph-19-13192-t001:** Evaluation system of the digital economy.

Target Level	Criterion Level	Index Level	Unit	Indicator Direction
Digital Economy	Basic indicators	Number of IPV4 addresses	10,000	+
		Base stations of mobile phones	CNY 10,000	+
		Length of optical cable lines	km	+
		Number of internet broadband subscriber ports	10,000 ports	+
		Popularisation rate of mobile phones	%	+
	Industry indicators	Business volume of telecommunications services	CNY 100 million	+
		Income from related software business	CNY 10,000	+
		Number of top 100 internet companies	unit	+
	Environmental indicators	Number of software developers	unit	+
		Number of invention patent applications	piece	+
		Full-time equivalent of R&D personnel	10,000 man-years	+
		Transaction value in technical market	CNY 100 million	+
	Convergence indicators	Sales and purchases through e-commerce	CNY 100 million	+
		The proportion with e-commerce transactions enterprises	%	+
		Digital financial inclusion index	/	+

**Table 2 ijerph-19-13192-t002:** Results of measuring the development level of the digital economy.

Region		2016	2017	2018	2019	2020	Average Annual Growth Rate (%)
Eastern Region	Beijing	0.4123	0.5447	0.5977	0.6451	0.7380	16.04
	Tianjin	0.0670	0.0682	0.0796	0.0927	0.1047	11.98
	Hebei	0.0671	0.0884	0.1037	0.1208	0.1366	19.63
	Shanghai	0.2562	0.2686	0.3089	0.3385	0.3558	8.64
	Jiangsu	0.3704	0.3880	0.4389	0.4540	0.5106	8.45
	Zhejiang	0.2667	0.2884	0.3326	0.3636	0.4003	10.72
	Fujian	0.1395	0.1571	0.1849	0.1946	0.1848	7.64
	Shandong	0.2089	0.2436	0.3036	0.3179	0.3512	14.11
	Guangdong	0.4786	0.5292	0.6208	0.6893	0.7465	11.80
	Hainan	0.0289	0.0353	0.0360	0.0406	0.0427	10.59
	Average	0.2296	0.2612	0.3007	0.3257	0.3571	11.72
Central Region	Shanxi	0.0369	0.0467	0.0577	0.0653	0.0727	18.62
	Anhui	0.1035	0.1199	0.1502	0.1530	0.1759	14.49
	Jiangxi	0.0412	0.0634	0.0743	0.0933	0.1056	27.48
	Henan	0.1001	0.1095	0.1318	0.1418	0.1559	11.82
	Hubei	0.1261	0.1514	0.1858	0.2033	0.2130	14.23
	Hunan	0.0861	0.1029	0.1193	0.1431	0.1630	17.31
	Average	0.0823	0.0990	0.1198	0.1333	0.1477	15.83
Western Region	Mongolia	0.0298	0.0389	0.0431	0.0540	0.0633	20.92
	Guangxi	0.0524	0.0589	0.0715	0.0834	0.0981	17.00
	Chongqing	0.0773	0.0852	0.1138	0.1182	0.1244	13.21
	Sichuan	0.1196	0.1581	0.1898	0.2153	0.2374	19.00
	Guizhou	0.0393	0.0514	0.0648	0.0795	0.0884	22.68
	Yunnan	0.0562	0.0530	0.0670	0.0826	0.0953	14.85
	Shaanxi	0.0950	0.1264	0.1345	0.1578	0.1530	13.44
	Gansu	0.0280	0.0321	0.0405	0.0479	0.0543	18.05
	Qinghai	0.0135	0.0159	0.0212	0.0234	0.0274	19.70
	Ningxia	0.0163	0.0195	0.0244	0.0265	0.0291	15.74
	Xinjiang	0.0240	0.0286	0.0416	0.0478	0.0565	24.46
	Average	0.0501	0.0607	0.0738	0.0851	0.0934	16.93
Northeast Region	Liaoning	0.1099	0.1398	0.1287	0.1439	0.1378	06.71
	Jilin	0.0303	0.0414	0.0542	0.0555	0.0619	20.32
	Heilongjiang	0.0411	0.0508	0.0509	0.0601	0.0680	13.77
	Average	0.0604	0.0773	0.0779	0.0865	0.0892	10.72
National	Average	0.1174	0.1368	0.1591	0.1751	0.1917	13.09

**Table 3 ijerph-19-13192-t003:** Manufacturing transformation efficiency evaluation index.

Target Level	Criterion Level	Index Level	Unit	Indicator Direction
Manufacturing transformation efficiency	Economic benefits	Total profits of industrial enterprises above designated size_Manufacturing	CNY 100 million	+
	Green development	Energy consumption per unit of industrial added value	10,000 tons of standard coal/CNY 100 million	−
		Total industrial wastewater discharge	ton	−
		Common industrial solid wastes generated	10,000 tons	−
		Total industrial waste gas emissions	ton	−
	Technology innovation	R&D expenditure of industrial enterprises above designated size	CNY 10,000	+
		Number of valid invention patents for industrial enterprises above designated size	piece	+
	Digital convergence	Digital economy development level	/	+

**Table 4 ijerph-19-13192-t004:** Results of green and low-carbon transition efficiency measurement of manufacturing.

Region		2016	2017	2018	2019	2020	Average Annual Growth Rate (%)
Eastern Region	Beijing	0.1839	0.2210	0.2258	0.2395	0.2610	9.36
	Tianjin	0.0678	0.0624	0.0651	0.0638	0.0684	0.40
	Hebei	0.0420	0.0516	0.0591	0.0669	0.0770	16.41
	Shanghai	0.1442	0.1593	0.1818	0.1902	0.2143	10.48
	Jiangsu	0.1695	0.1907	0.2160	0.2351	0.2668	12.03
	Zhejiang	0.1300	0.1430	0.1594	0.1706	0.1911	10.12
	Fujian	0.0643	0.0757	0.0836	0.0887	0.0978	11.14
	Shandong	0.6556	0.5603	0.4499	0.3293	0.4136	−8.87
	Guangdong	0.1790	0.1996	0.2261	0.2484	0.2716	10.10
	Hainan	0.0503	0.0553	0.0587	0.0617	0.0652	6.73
	Average	0.1687	0.1719	0.1726	0.1694	0.1927	3.56
Central Region	Shanxi	0.0266	0.0540	0.0688	0.0592	0.0675	32.56
	Anhui	0.0584	0.0714	0.0820	0.0881	0.1043	15.73
	Jiangxi	0.0517	0.0646	0.0736	0.0816	0.0938	16.19
	Henan	0.0671	0.0764	0.0863	0.0948	0.1080	12.65
	Hubei	0.0757	0.0840	0.0955	0.1035	0.1087	9.52
	Hunan	0.1578	0.1892	0.2130	0.2416	0.2775	15.19
	Average	0.0729	0.0899	0.1032	0.1115	0.1266	14.94
Western Region	Mongolia	0.0538	0.0566	0.0569	0.0589	0.0647	4.75
	Guangxi	0.0335	0.0347	0.0367	0.0385	0.0434	6.79
	Chongqing	0.0603	0.0675	0.0752	0.0798	0.0857	9.20
	Sichuan	0.0430	0.0507	0.0564	0.0615	0.0687	12.51
	Guizhou	0.0319	0.0359	0.0391	0.0457	0.0487	11.26
	Yunnan	0.0420	0.0466	0.0504	0.0552	0.0618	10.13
	Shaanxi	0.0394	0.0460	0.0487	0.0524	0.0533	8.03
	Gansu	0.0253	0.0272	0.0289	0.0301	0.0347	8.23
	Qinghai	0.0376	0.0377	0.0394	0.0429	0.0457	5.02
	Ningxia	0.0288	0.0304	0.0322	0.0333	0.0354	5.27
	Xinjiang	0.0370	0.0435	0.0487	0.0523	0.0623	14.03
	Average	0.0393	0.0434	0.0466	0.0500	0.0550	8.73
Northeast Region	Liaoning	0.0445	0.0507	0.0519	0.0546	0.0624	8.98
	Jilin	0.0366	0.0380	0.0392	0.0408	0.0439	4.70
	Heilongjiang	0.0495	0.0542	0.0554	0.0605	0.0678	8.24
	Average	0.0435	0.0476	0.0488	0.0520	0.0580	7.53
National	Average	0.0896	0.0960	0.1001	0.1023	0.1155	6.64

**Table 5 ijerph-19-13192-t005:** Variables and explanations.

Variable	Symbol	Type	Measurement Method	Unit
Green and low-carbon transformation efficiency in manufacturing	$MTE_{it}$	Explained variables	Economic benefits, green development, technological innovation, and digital integration	/
Digital economy development level	$DE_{it}$	Core explanatory variables	Basic indicators, industrial indicators, environmental indicators, and integration indicators are composed	/
Technology innovation	$RD_{it}$	Intermediate variable	R&D expenditure of industrial enterprises above designated size	CNY 100 million
Human capital	$LAB_{it}$	Control variables	Average number of years of schooling index for people over 6 years old	person/year
Operating costs	$COST_{it}$		The main business cost of industrial enterprises above designated size	CNY 100 million
Industry scale	$SCA_{it}$		Industrial value added as a percentage of GDP	%

**Table 6 ijerph-19-13192-t006:** Statistical description of the variables.

Variables	Observations	Mean	Median	Std. Dev.	Minimum	Maximum
$MTE_{it}$	150	−2.583	−2.774	0.697	−3.675	−0.422
$DE_{it}$	150	−2.278	−2.288	0.917	−4.309	−0.292
$RD_{it}$	150	14.53	14.80	1.373	11.12	17.03
$SCA_{it}$	150	−1.218	−1.149	0.857	−3.803	0.707
$COST_{it}$	150	9.862	9.882	1.055	7.175	11.81
$LAB_{it}$	150	2.231	2.227	0.0930	2.045	2.548

**Table 7 ijerph-19-13192-t007:** Benchmark regression.

Variables	(1)	(2)
	$MTE_{it}$	$MTE_{it}$
$DE_{it}$	0.559 *** (0.041)	0.477 *** (0.055)
$SCA_{it}$		−0.081 (0.211)
$COST_{it}$		0.314 *** (0.068)
$LAB_{it}$		1.555 *** (0.578)
Constant		−8.164 *** (1.535)
obs	150	150
$R^{2}$	0.606	0.689
$R^{2}\_a$	0.507	0.600

Note: *** represents 1% significance level.

**Table 8 ijerph-19-13192-t008:** Mediation effect test result.

Variables	(1)	(2)	(3)
	$MTE_{it}$	$RD_{it}$	$MTE_{it}$
$DE_{it}$	0.452 *** (0.0419)	0.516 *** (0.0518)	0.331 *** (0.0505)
$RD_{it}$			0.246 *** (0.0637)
$SCA_{it}$	−0.292 *** (0.0797)	0.0160 (0.0727)	−0.303 *** (0.0816)
$COST_{it}$	0.289 *** (0.0527)	0.756 *** (0.0579)	0.143 ** (0.0642)
$LAB_{it}$	1.119 ** (0.478)	0.744 (0.527)	0.822 * (0.472)
Constant	−7.252 *** (1.251)	6.619 *** (1.407)	−9.022 *** (1.277)
Mediating effect	Presence		
Mediating effect percentage	0.281		
obs	150	150	150

Note: *** *p* < 0.01, ** *p* < 0.05, and * *p* < 0.1.

**Table 9 ijerph-19-13192-t009:** Heterogeneity test result.

Variables	Eastern Region	Central Region	Western Region	Northeast Region
$DE_{it}$	0.684 *** (0.182)	0.806 *** (0.164)	0.340 *** (0.032)	0.392 *** (0.109)
$SCA_{it}$	0.910 (0.580)	0.013 (0.493)	−0.546 *** (0.138)	−0.584 (0.443)
$COST_{it}$	0.545 *** (0.134)	0.162 (0.202)	0.154 *** (0.053)	0.282 *** (0.082)
$LAB_{it}$	1.881 * (1.034)	1.149 (2.402)	0.830 ** (0.330)	1.360 (1.333)
Constant	−9.844 *** (3.048)	−4.815 (6.526)	−5.979 *** (0.933)	−8.457 ** (2.912)
obs	50.000	30.000	55.000	15.000
$R^{2}$	0.659	0.830	0.923	0.882
$R^{2}\_a$	0.536	0.754	0.896	0.794

Note: ***, **, * represent 1%, 5%, 10% significant levels respectively.

**Table 10 ijerph-19-13192-t010:** Robustness test result.

Variable	(1)	(2)–(5)				(6)	(7)	(8)
$DE_{it}$	1.239 *** (0.108)	0.559 *** (0.041)	0.587 *** (0.054)	0.546 *** (0.050)	0.477 *** (0.055)	0.522 *** (0.071)	0.329 *** (0.055)	0.881 *** (0.213)
$SCA_{it}$	0.350 (0.255)		0.177 (0.221)	−0.142 (0.215)	−0.081 (0.211)	0.018 (0.309)	−0.275 (0.221)	−0.310 *** (0.038)
$COST_{it}$	0.275 *** (0.077)			0.323 *** (0.070)	0.314 *** (0.068)	0.271 *** (0.086)	0.213 *** (0.074)	−0.013 (0.137)
$LAB_{it}$	1.584 ** (0.677)				1.555 *** (0.578)	1.733 ** (0.785)	1.245 ** (0.519)	−1.006 (0.865)
Constant	−15.436 *** (1.457)	−1.309 *** (0.095)	−1.030 *** (0.360)	−4.701 *** (0.857)	−8.164 *** (1.535)	−7.865 *** (2.125)	−6.985 *** (1.555)	1.421 (3.605)
obs	150	150	150	150	150	120	120	150
$R^{2}$	0.786	0.606	0.608	0.669	0.689	0.615	0.639	0.707
$R^{2}\_a$	0.725	0.507	0.505	0.579	0.600	0.467	0.500	0.698

Note: *** and ** represent 1% and 5% significance levels, respectively.

## Data Availability

The data involved in this study are all from public data.

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
