# Peer review of "An Empirical Analysis of the Impact of Digital Economy on Manufacturing Green and Low-Carbon Transformation under the Dual-Carbon Background in China"

_ijerph, 2022, doi:10.3390/ijerph192013192_

Round 1

Reviewer 1 Report

The authors can improve the analysis of the data.

(a) Add more comments on the results shown in Table 6. 

(b) Although "Regression coefficients of the digital economy are positive and significant at the 1% significance level with or without control variables", it is unclear how strongly the regressors correlate with each other. A multicollinearity analysis can be helpful in this study. 

(c) What happens if we remove SCA from model 2 in table 7? 

(d) If present, serial correlation in panel-data models biases the standard errors and causes the results to be less efficient. Are the data free of time correlation? If not, I suggest the weighted least squares approach.

(e) The p-values in Tables 7-9 may be meaningless if the data is not Gaussian or the sample size is not sufficiently large.

Reviewer 2 Report

Three areas, all relatively minor, should be addressed before publication.  

First, some jargon words need to be better defined.  For example the development model of 181 the ʺthree highsʺ and the ʺdouble carbonʺ  goal are mentioned and put in quotes in the text.  But none of them are defined adequately for an uninformed reader. 

For the DE index, Indicators are all in terms of raw numbers.  But does this lead to a bias by simple largeness of the urban area.  A massive urban center will have more of these things even if their interpenetration in the local economy is much less extensive than a smaller urban area.  It’s not entirely clear how the different indicators then are transformed into the index shown on pages 6-7 for each province.  More detail about this would be good. Perhaps an example of how the various raw measures lead into the computation of the index value for a province would be helpful here.  

More clarity is also needed concerning the manufacturing efficiency index.  For instance, it looks as if some raw emissions levels are part of the index, while others are ratios per added value.  The overall index should be an efficiency measure I think.  It’s not currently clear if it is.  Again, this needs to be explained more effectively.  If it is as it should be (the more detailed explanation will make this clear) then all is good.  But currently there is insufficient detail in the paper to be able to tell if it is in fact measuring what it purports to measure.  Perhaps an example of how the various raw measures lead into the computation of the index value for a province would be helpful here.

Reviewer 3 Report

This paper has the potential to contribute to the literature. I suggest authors address the following issues:

1. Improve the research problem in the Introduction section.

2. Explain the tests you have conducted to detect the type of endogeneity issues and then justify the selection of methods applied to correct the endogeneity problem.

3. Compare your results from past papers and link the theory in the discussion of results.

4. Improve practical implications of your study findings.

Reviewer 4 Report

The paper concerns impact of digital economy on green transformation, The paper is quite original and presents discussion on results of green manufacturing and green economy.

In the first sentence of the abstract authors present hypothesis of their research work. I wonder if this sentence is from other authors’ book or it is the research hypothesis in this research, please, explain.

I noticed you have repeated this sentence many times in your paper, and you  provide general expanations.

“(1) Research on the digital economy. The term ʺdigital economyʺ was first introduced” 51 row – please, explain why do you apply these punctuation.

“However, there is a lack of research on the digital economy in terms of the green” 116 row please explain, what are the fundamentals of this conclusion.

“It can be seen that the digital economy development level has significant heterogeneity in time and space” – I’m not sure if it is correct I would say that development is various and heterogeneous.

Please, add which method has been used for the trends’ estimation,

„Hypothesis H2 is verified.” – Where is hypothesis H1, and how many hypotheses have you formulated, why?  what variables have been taken in hypotheses and  why?

In this study you verify technology generally but not about “effect of the digital economy, accelerate the application of emerging technologies such as robotics, big data, and 5G, focus on building digital infrastructure represented by artificial 597

intelligence, industrial internet, 5G base stations, and big data centres, accelerate the cross‐ 598

fertilisation of new‐generation internet information technology with the manufacturing 599

industry, guide enterprises to orderly and reasonably apply digital technology in produc‐ 600

tion activities, and promote industrial digitization and informatization”

Round 2

Reviewer 1 Report

No further comments.